# Planetary Gearbox Fault Diagnosis Based on ICEEMD-Time-Frequency Information Entropy and VPMCD

**Yihan Wang [1,2]**, **Zhonghui Fan [1,2]**, **Hongmei Liu [3,4,\*]** **and Xin Gao [1,2]**

[1] CEPREI, Guangzhou 510610, China; zy1714208@buaa.edu.cn (Y.W.); 18620809907@163.com (Z.F.); gaoxin@ceprei.com (X.G.)

[2] Key laboratory of Industrial Equipment Quality Big Data, MIIT, Guangzhou 510610, China

[3] School of Reliability and Systems Engineering, Beihang University, Beijing 100191, China

[4] Science & Technology Laboratory on Reliability & Environmental Engineering, Beihang University, Beijing 100191, China

\* Correspondence: liuhongmei@buaa.edu.cn

**Abstract:** Planetary gearboxes are more and more widely used in large and complex construction machinery such as those used in aviation, aerospace fields, and so on. However, the movement of the gear is a typical complex motion and is often under variable conditions in real environments, which may make vibration signals of planetary gearboxes nonlinear and nonstationary. It is more difficult and complex to achieve fault diagnosis than to fix the axis gearboxes effectively. A fault diagnosis method for planetary gearboxes based on improved complementary ensemble empirical mode decomposition (ICEEMD)-time-frequency information entropy and variable predictive model-based class discriminate (VPMCD) is proposed in this paper. First, the vibration signal of planetary gearboxes is decomposed into several intrinsic mode functions (IMFs) by using the ICEEMD algorithm, which is used to determine the noise component by using the magnitude of the entropy and to remove the noise components. Then, the time-frequency information entropy of intrinsic modal function under the new decomposition is calculated and regarded as the characteristic matrix. Finally, the fault mode is classified by the VPMCD method. The experimental results demonstrate that the method proposed in this paper can not only solve the fault diagnosis of planetary gearboxes under different operation conditions, but can also be used for fault diagnosis under variable operation conditions. Simultaneously, the proposed method is superior to the wavelet entropy method and variational mode decomposition (VMD)-time-frequency information entropy.

**Keywords:** planetary gearbox; ICEEMD; time-frequency information entropy; VPMCD; fault diagnosis

---

## 1. Introduction

As the key unit of a transmission device, planetary gearboxes have the advantages of a compact structure, large transmission moment, and accurate transmission ratio, and so on [1,2]. Once the planetary gearbox is damaged, it will cause machine breakdowns. In severe cases, it will lead to huge economic losses [3,4]. What is more, planetary gearboxes are often operated under heavy loads, which are very prone to failure [5]. Therefore, fault diagnosis of planetary gearboxes is significant to production safety and cost efficiency [6]. Vibration signal analysis method is one of the key tools for planetary gearbox fault diagnosis [7]. The vibration of planetary gearboxes are complex and very easy to be drowned out by noise, which makes the weak features of incipient faults difficult to detect [8]. Traditional vibration signal analysis methods such as time domain feature analysis and frequency

domain analysis do not reflect the fault information of planetary gearboxes and the diagnostic accuracy of fault diagnosis is unsatisfactory.

For this reason, time-frequency analysis has emerged as an important signal processing method and has a wide range of applications in the field of vibration signals. Many time-frequency methods, such as wavelet transform and empirical mode decomposition (EMD), are applied to analyze nonlinear and nonstationary signals of planetary gearboxes [9]. However, the basis function of wavelet transform has no self-adaptability. It needs to be a preselected fixed-basis function according to the characteristics of the different signals. The empirical mode decomposition algorithm can adaptively decompose the vibration signal into a series of intrinsic mode functions (IMFs) from high frequency to low frequency [10], but usually generates problem of mode mixing. Later, a series of derived methods came into being such as ensemble empirical mode decomposition (EEMD), local characteristic-scale decomposition (LCD), and complementary ensemble empirical mode decomposition (CEEMD), and so on. However, low-speed and heavy-duty operation conditions with high noise environments frequently cause faults in the planetary gearboxes. It is more difficult to diagnose a fault than to fix the axis gearboxes, in which accurate diagnosis is very important. Due to the interference of noise, the problem of mode mixing is more severe, which leads to difficulty in completing the subsequent feature extraction. The above methods do not solve the problem caused by high noise. It is necessary to find a new method of signal features analysis with noise reduction to suppress the influence of noise on the vibration signal. The permutation entropy algorithm, a useful tool in amplifying slight changes of signals, has advantages of generality and easy computing in nonlinear and nonstationary signals [11]. Based on the permutation entropy, Aziz et al. proposed multi-scale permutation entropy (MPE) to depict tiny changes of signals from different local scales rather than a single scale [12]. Therefore, the MPE algorithm was utilized to effectively distinguish components between noise and original vibration signals of planetary gearboxes. For the reasons given above, in this paper, a method of improved complementary ensemble empirical mode decomposition (ICEEMD) is proposed for the first time.

In terms of feature extraction, time-frequency information entropy is utilized to reveal the complexity hidden in signals and can effectively reflect fault information of time-frequency characteristics. Thus, time-frequency information entropy is regarded as a feature characteristic parameter and is applied to realize the robust feature extraction [13].

For mechanical fault diagnosis, different fault classification methods directly affect the efficiency and accuracy of fault recognition. The algorithm of variable predictive model-based class discriminate (VPMCD) makes use of the inherent correlation of sample eigenvalues to establish a feature learning model [14], which can be effectively applied to multi-classification mechanical fault diagnosis with small samples [15]. In addition, the VPMCD algorithm avoids the problem of over-fitting of neural network classification and kernel function selection of SVM (Support Vector Machine) classification. At the same time, the computing time is reduced and the robustness of the algorithm is improved [16].

In the study presented in this paper, a fault diagnosis method of planetary gearboxes based on ICEEMD-time-frequency information entropy and VPMCD is first proposed. First, vibration signals of planetary gearboxes are collected by the accelerometer. Second, several IMFs are obtained by using the method of ICEEMD. The method proposed can eliminate noise on the basis of each IMF, which avoids the disadvantages of loss of feature information caused by overall noise reduction. Afterwards, each IMF is calculated by time-frequency information entropy as feature vectors. Then, by the method of principal component analysis (PCA), the dimension of feature vectors is reduced. In order to verify the effectiveness of the proposed method, the diagnostic performance of other two methods under various working conditions is also investigated in fault feature extraction. Compared with the other two methods, the ICEEMD-time-frequency information entropy method proposed in this paper has an advantage for feature extraction and allows the fault features to be effectively distinguished in the feature space. Finally, the VPMCD model is utilized to classify different fault modes.

This paper is organized as follows: Section 2 introduces the relevant feature extraction methodology, which includes the ICEEMD, time-frequency information entropy, and principal component analysis

(PCA); Section 3 presents the principle of classification about VPMCD. In Section 4, the scheme of the planetary gearbox based on ICEEMD-time-frequency information entropy and VPMCD are described. Section 5 describes the case study to validate the entire method; Conclusions are drawn in Section 6.

## 2. Feature Extraction Method Based on ICEEMD-Time-Frequency Information Entropy

Vibration signals of planetary gearboxes are complex, with multiple components and nonlinear and nonstationary characteristics that are obviously interfered with by noise [17]. How to obtain fault characteristic information effectively is an important part of fault diagnosis of planetary gearboxes. In this study, the vibration signal feature is extracted by ICEEMD-time-frequency information entropy to characterize the operating state of the planetary gearbox.

### 2.1. A Description of Improved Complementary Ensemble Empirical Mode Decomposition (ICEEMD)

The vibration signal of planetary gearboxes, with multiple components and nonlinear, nonstationary, and strong time-varying characteristics, is complex and easily buried by noise. However, traditional fault diagnosis methods fail to effectively diagnose the faults of planetary gearboxes. It is essential to find out a suitable method to suppress interference components in order to obtain intrinsic features of the vibration signal.

Comparing complementary ensemble empirical mode decomposition (CEEMD) with ensemble empirical mode decomposition (EEMD) and empirical mode decomposition (EMD) [18], CEEMD is an improved algorithm that reduces the mode mixing problem [19]. However, the reconstructed components of CEEMD contain noise components that cannot be eliminated. In order to eliminate the influence of noise on feature extraction, we used multi-scale permutation entropy with signal noise reduction for the first time. Multi-scale permutation entropy combines multi-scale entropy with permutation entropy to analyze sequence information more effectively [20–22], which can measure noise characteristics from multiple scales rather than a single scale. Using this method, the noise components can be accurately identified. In this paper, the improved complementary ensemble empirical mode decomposition algorithm (ICEEMD) was used to analyze the vibration signal of planetary gearboxes for the first time. The specific ICEEMD algorithm steps are shown in Figure 1.

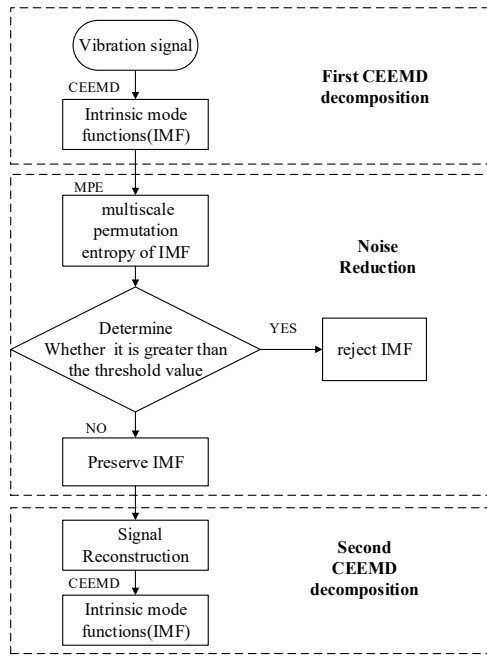

**Figure 1.** Algorithm flowchart of improved complementary ensemble empirical mode decomposition (ICEEMD).

Step 1: Use CEEMD to decompose the original signal.

(1) The original signal is $S(t)$. Add white noise signals $n_i(t)$ and $-n_i(t)$ with the mean value of zero onto the original signal. Then, the signal after adding white noise is obtained:

$$S_i^+(t) = S(t) + a_i n_i(t) \tag{1}$$

$$S_i^-(t) = S(t) - a_i n_i(t) \tag{2}$$

where $a_i$ is the amplitude of the white noise, $i = 1, 2, \ldots, Ne$, and $Ne$ is the logarithm of the added white noise.

(2) Signal $S_i^+(t)$ and signal $S_i^-(t)$ are decomposed by EMD to obtain $IMF_{i1}^+$ and $IMF_{i1}^-$. Then, the integrated average value of $IMF_{i1}^+$ and $IMF_{i1}^-$ are obtained by Equation (3).

$$IMF_1 = \frac{1}{2Ne} \sum_{i=1}^{Ne} \left[ IMF_{i1}^+ + IMF_{i1}^- \right] \tag{3}$$

(3) Repeat the above steps until the termination condition of EMD is met.

Step 2: Reduce noise by using the MPE algorithm.

The basic idea of multi-scale permutation entropy is the coarse graining of the original signal [23]. The specific steps for calculating the multi-scale arrangement entropy of IMF are as follows:

(1) Coarse graining of IMF.

$IMF_p = \{x_i, i = 1, 2, \ldots, N\}, 1 \le p \le M, p$ is the number of IMFs.
Coarse graining sequence:

$$y_j^s = \frac{1}{s} \sum_{i=(j-1)s+1}^{js} x_i, \ j = 1, 2, \ldots \frac{N}{s} \text{ and } j \text{ is an integer.} \tag{4}$$

where $s$ is the scale factor.

(2) Time reconstruction of coarse graining sequence $y_j^s$:

$$Y_l^s = \left\{ y_l^s, y_{l+\tau}^s, \ldots, y_{l+(m-1)\tau}^s \right\}$$

where $m$ is embedding dimension, $\tau$ is delay time, and $l$ is a reconstructed component, where the expression for $l$ is $l = 1, 2, \ldots, N - (m-1)\tau$.

Arrange the $Y_l^s$ in ascending order and calculate the probability $P_r$ of each sequence, where $r = 1, 2, \ldots, R, R \le m!$

Calculate the multi-scale permutation entropy of each coarse graining sequence.

$$IMF_P(m) = - \sum_{r=1}^{R \sum_r} P_r ln \tag{5}$$

(3) Eliminate noise components.

The multi-scale permutation entropy of each IMF component is calculated and compared with the set threshold value. Then, eliminate noise components larger than the threshold value.

Step 3: Second CEEMD decomposition.

Reconstruct signal. The reconstructed signal $X(t)$ is decomposed by CEEMD to obtain new intrinsic modal components (IMFs). Refer to step 1 for detailed steps.

In order to test the advantages of the method proposed in this paper, the ICEEMD method and CEEMD method were used to decompose the simulation signal, where $Y$ is the simulation signal, $x_n$ is the noise signal, and S is the simulation signal with noise.

$$Y = 2.5sin(2\pi 8t + \pi/6) + 3tsin(2\pi 4t + \pi/2) \tag{6}$$

The simulation signal, noise signal, and simulation signal with noise are shown in Figure 2.

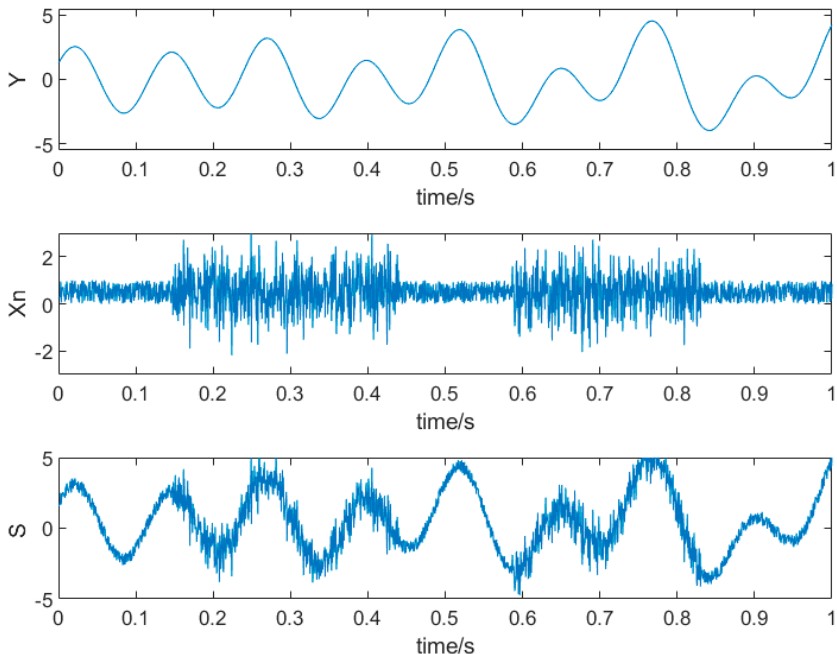

**Figure 2.** Simulation signal, noise signal, and simulation signal with noise.

The seven sub-signals decomposed by ICEEMD are shown in Figure 3 and the seven sub-signals decomposed by CEEMD are shown in Figure 4.

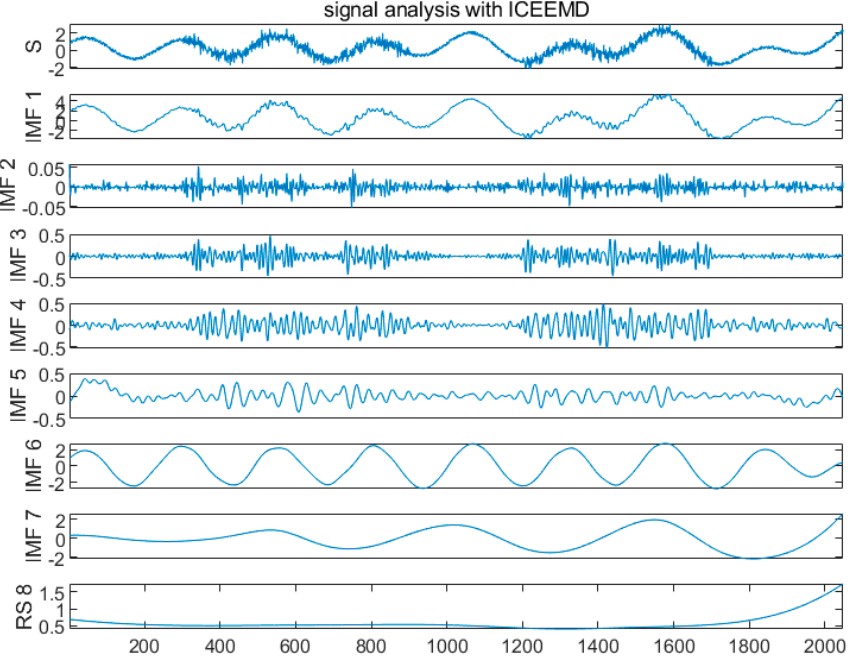

**Figure 3.** Seven sub-signals decomposed by ICEEMD.

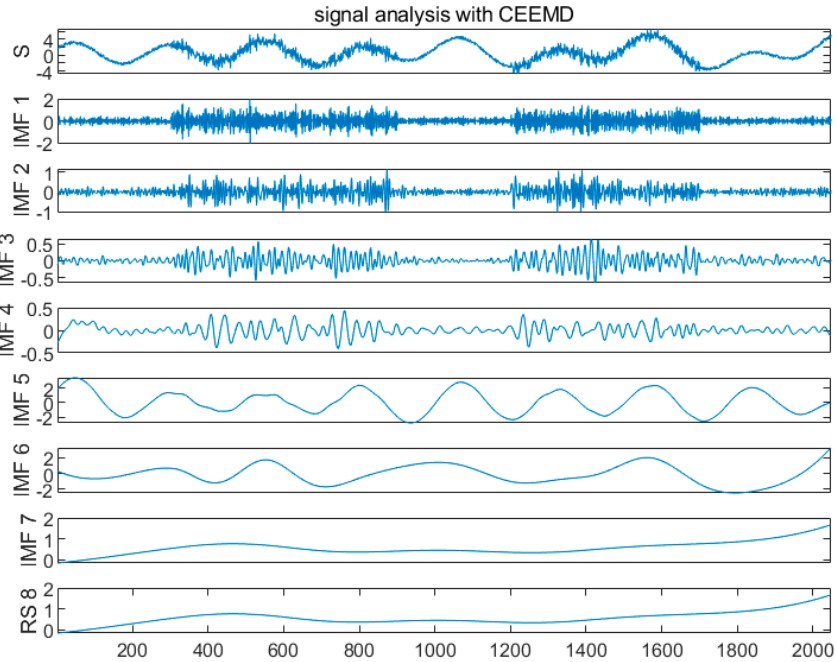

**Figure 4.** Seven sub-signals decomposed by complementary ensemble empirical mode decomposition (CEEMD).

As shown in Figures 3 and 4, seven IMFs are sequentially decomposed in order of their frequency using the two methods. We can see that the decomposed components of CEEMD are more severely affected by noise than those of ICEEMD. In order to further test the performance of the two methods in eliminating noise, the simulation signal was compared with the reconstructed signals of ICEEMD and CEEMD, respectively, and the results were shown in Table 1.

**Table 1.** The comparison of ICEEMD and CEEMD.

| Method | MSE (Mean Squared Error) | IO (Index of Orthogonality) | R (Coefficient of Determination) |
|---|---|---|---|
| **ICEEMD** | 4.0286 | 0.9073 | 0.9925 |
| **CEEMD** | 6.2469 | 1.3127 | 0.9489 |

By comparison, the correlation between the simulation signal and the reconstructed signal of ICEEMD is higher and the mean squared error is smaller than that of CEEMD. In addition, compared with CEEMD, the components of the ICEEMD method have better orthogonality. Therefore, the method proposed in this paper is better at removing noise and restoring the signal characteristics.

*2.2. A Description of Time-Frequency Information Entropy*

Time-frequency distribution of the signal reflects its energy variation at each frequency. It can quantitatively describe the different degrees of planetary gearboxes under different operation conditions. Time-frequency information entropy can describe the time-frequency distribution complexity of signals, so it is especially suitable for fault feature extraction. In this paper, time-frequency information entropy is the feature vector for fault diagnosis. It can represent the fault information hidden in the signal, with high diagnostic accuracy and good robustness.

The steps of time-frequency information entropy can be described as follows [24,25]:

(1)　Short-time Fourier transform for each IMF component.

$$F(\text{t},\omega) = \int IMF(i)w(i-t)e^{-j\omega t}d\iota \qquad (7)$$

where $w(\tau)$ is the window function.

(2)   Calculate the time-frequency energy spectrum of each IMF.

The time-frequency energy spectrum is used to describe the time-frequency distribution of the signal. The expression is as follows [26]:

$$S(t,\omega) = \left|F(t,\omega)\right|^2 \tag{8}$$

(3)   Energy normalization.

The difference of energy distribution of time-frequency block on time-frequency plane reflects the difference in signal time-frequency distribution. The time-frequency energy $W_i$ of $IMF_i$ is obtained by the Equations (6) and (7). The time-frequency energy of all IMFs is A.

Energy normalization of $IMF_i$:

$$q_i = W_i/A(i = 1,\ldots,M) \tag{9}$$

(4)   Calculate the entropy value of each time-frequency block:

$$s(q) = -\sum_{i=1}^{M} q_i \ln q_i \tag{10}$$

## 3. Fault Classification Based on VPMCD

In mechanical fault diagnosis, there is an intrinsic relationship between the characteristic values of signals. Their intrinsic relationships will change differently under different operation conditions [27–29].

The variable predictive model based class discriminate (VPMCD) method is a pattern recognition method based on the variable predictive model (VPM). According to the relationship between characteristic values, a prediction model is established to classify the test samples [30–33]:

Linear model (L):

$$X_i = b_0 + \sum_{j=1}^{r} b_j X_j \tag{11}$$

Linear interaction model (LI):

$$X_i = b_0 + \sum_{j=1}^{r} b_j X_j + \sum_{j=1}^{r} \sum_{k=j+1}^{r} b_{jk} X_j X_k \tag{12}$$

Quadratic interaction model (QI):

$$X_i = b_0 + \sum_{j=1}^{r} b_j X_j + \sum_{j=1}^{r} b_{jj} X_j^2 + \sum_{j=1}^{r-1} \sum_{k=j+1}^{r} b_{jk} X_j X_k \tag{13}$$

Quadratic model (Q):

$$X_i = b_0 + \sum_{j=1}^{r} b_j X_j + \sum_{j=1}^{r} b_{jj} X_j^2 \tag{14}$$

where $r$ is model order, $r \le k - 1$, $k$ is number of feature values for each fault type, and $j$ is number of fault types. $X_j(j \ne i)$ is the predictive variable, $X_i$ is the predicted variable, and $b_0, b_j, b_{jj}$ and $b_{jk}$ are model parameters.

Select one of the four models as a predictive model, and its expression is as follows [34,35]:

$$X_i = f\left(X_j, b_0, b_j, b_{jj}, b_{jk}\right) + e \tag{15}$$

where *e* is the prediction error.

Suppose there are *n* fault types, and *k* prediction models are built for each fault type to obtain $n \times k$ prediction models. Calculate the sum of squared prediction errors of *k* eigenvalues under the same fault type. The fault category corresponding to the model with the smallest sum of squares of the prediction error is the final output fault mode. Therefore, the VPMCD classification algorithm can be applied not only into linear classification, but also into nonlinear classification.

## 4. Fault Diagnosis Scheme for a Planetary Gearbox

This paper proposes a fault diagnosis method based on ICEEMD-time-frequency information entropy and VPMCD. The specific procedure of the diagnosis scheme is shown in Figure 5.

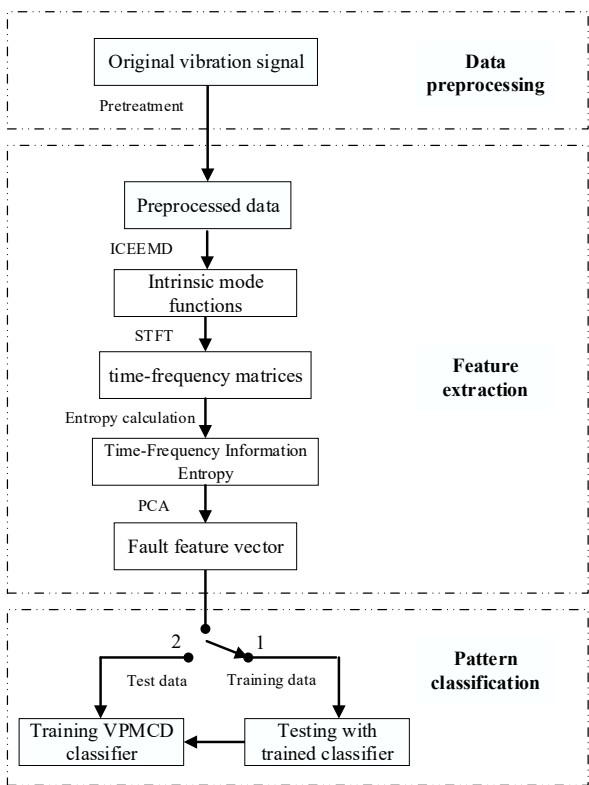

**Figure 5.** The fault diagnosis procedure for a planetary gearbox.

(1)    Preprocess the data of the original vibration signal.
(2)    Decompose the vibration signals of the planetary gearbox into a series of IMFs by utilizing ICEEMD.
(3)    Calculate the time-frequency information entropy of the IMFs as feature vectors by using short-time Fourier transform (STFT) and information entropy.
(4)    Reduce the dimension of the feature vectors by utilizing principal components analysis (PCA) to improve the accuracy and robustness of pattern recognition.
(5)    Classify the fault modes by using VPMCD.

## 5. Experimental Verification

Since the gear motion in the planetary gearbox is a typical composite motion, the vibration signal is more complex than that of a fixed shaft gearbox. Moreover, the gear is the most fault-prone part of the planetary gearbox. Therefore, this paper takes the gear of the planetary gearbox as the diagnostic object to verify the effectiveness of the proposed method. ICEEMD-time-frequency information entropy method was utilized to extract the fault feature of the gear vibration data from a fault-prediction test bed. We used the power transmission fault prediction test bed (DPS) manufactured by Spectra Quest,

USA. The test bed is shown in Figure 6, which includes a centrally fixed sun gear, a planetary carrier and ring gear, and four planetary gears that change with the center of rotation of the sun gear. In addition, the planetary gearbox of this experimental bench was small in volume, and the vibration caused by the fault was weak, which increases the difficulty of diagnosis. In this experiment, a total of 12 operation conditions were collected. The specific operation condition information is shown in Table 2. The signal sampling frequency was 12,800 Hz and the signal sampling point was 524,288. The experimental fault data includes five states: gear tooth crack fault, gear wear fault, tooth breaking fault, gearing missing fault, and normal. In this study, experimental data were divided into 40 segments, and each segment had 12,800 points.

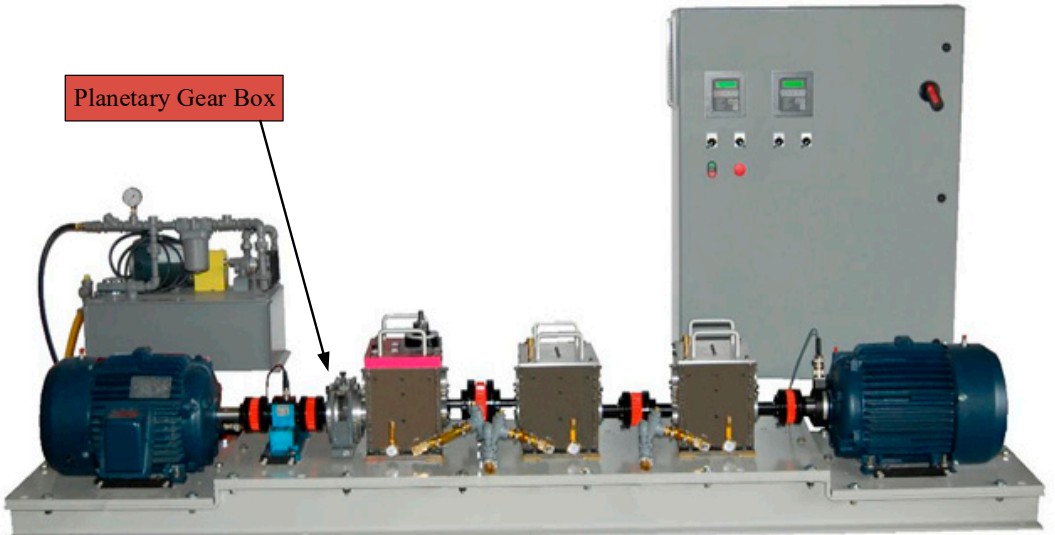

**Figure 6.** Test bed of planetary gearbox.

**Table 2.** The description of signal operation conditions.

| Operation Condition Number | Speed (Hz) | Load (Nm) |
| --- | --- | --- |
| Condition 1 | 20 | 0 |
| Condition 2 | 40 | 0 |
| Condition 3 | 60 | 0 |
| Condition 4 | 20 | 0.6 |
| Condition 5 | 40 | 0.6 |
| Condition 6 | 60 | 0.6 |
| Condition 7 | 10 | 1.2 |
| Condition 8 | 20 | 1.2 |
| Condition 9 | 30 | 1.2 |
| Condition 10 | 40 | 1.2 |
| Condition 11 | 50 | 1.2 |
| Condition 12 | 60 | 1.2 |

*5.1. Fault Feature Extraction Based on ICEEMD-Time-Frequency Information Entropy*

First of all, the vibration signal collected by the sensor was preprocessed.

Secondly, the ICEEMD algorithm was applied to decompose each fault mode. The threshold of multi-scale permutation entropy $\theta_0$ was 0.6, embedded dimension m was 6, time delay factor $\tau$ was 1, and the scale factor s was 5. The high frequency part of the vibration signal of the planetary gearbox contained the main fault signal. Therefore, the first six intrinsic modal components (IMFs) of the second CEEMD decomposition were taken.

Thirdly, the time-frequency distribution of IMF was computed by means of STFT. The time-frequency entropy of each failure mode was taken as the fault feature vector. The length of the time-frequency block was 64 and width was 64. The number of the lateral and longitudinal slip steps was equal to 32. Then, principal component analysis (PCA) was used to reduce the fault characteristics and obtain the 3-dimensional fault feature vector. The clustering results are shown in Figures 7–10 for the first four operation conditions.

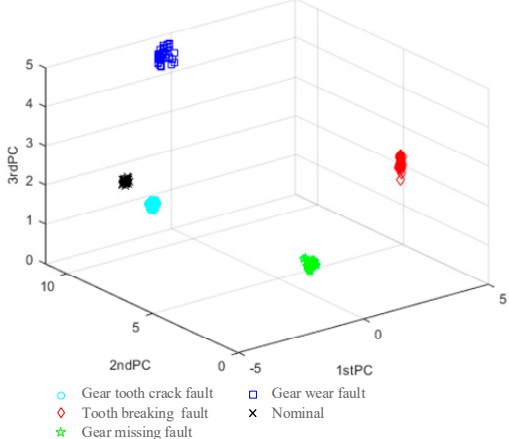

**Figure 7.** Clustering result of gear fault features under condition 1.

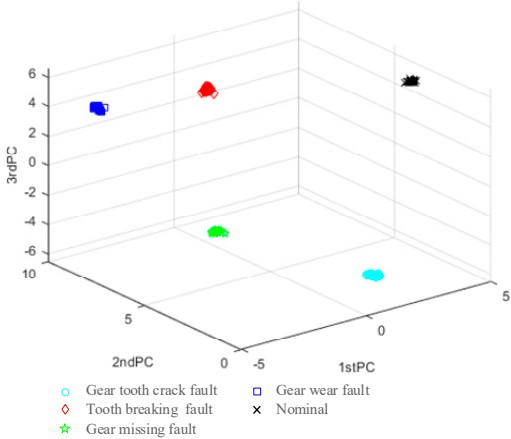

**Figure 8.** Clustering result of gear fault features under condition 2.

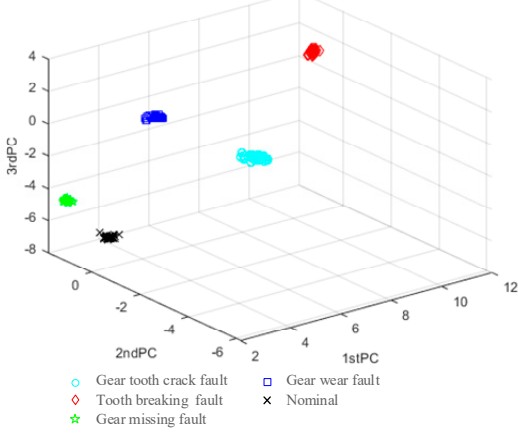

**Figure 9.** Clustering result of gear fault features under condition 3.

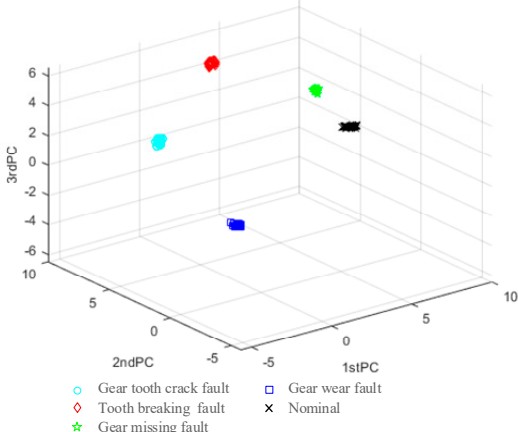

**Figure 10.** Clustering result of gear fault features under condition 4.

From Figures 7–10, it can be seen that different fault types can be well separated under the same operation condition. There is no feature vector mixing under different fault states. Moreover, the clustering effect about the eigenvalues of the same fault type shows satisfactory performance and no divergence.

*5.2. Method Comparison*

In order to reflect upon the effectiveness of the proposed algorithm, two feature extraction methods were used to compare with the proposed method in this paper.

5.2.1. Comparison between the Proposed Method with the Wavelet Entropy Method

Wavelet entropy, as a traditional time-frequency analysis method, is widely used in weak signal fault diagnosis. The specific algorithm steps are as follows:

First, the vibration signal is preprocessed.

Second, the planetary gearbox vibration signal is decomposed into three layers to obtain eight sub-band signals by using the sym6 wavelet base.

The third step is to calculate the total energy of each frequency band signal and normalize the energy. The wavelet entropy of each sub-band is calculated to obtain an 8-dimensional feature vector. The feature vector is dimension-reduced by principal component analysis (PCA). The clustering results are shown in Figures 11–14.

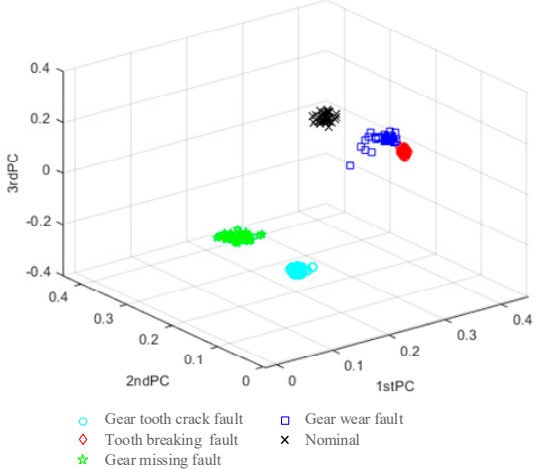

**Figure 11.** Clustering result of wavelet entropy under condition 1.

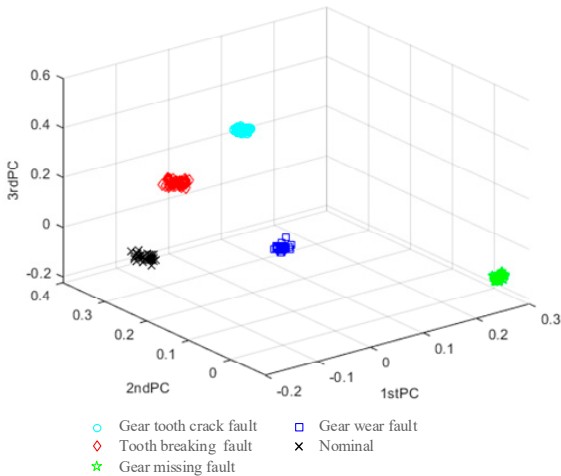

**Figure 12.** Clustering result of wavelet entropy under condition 2.

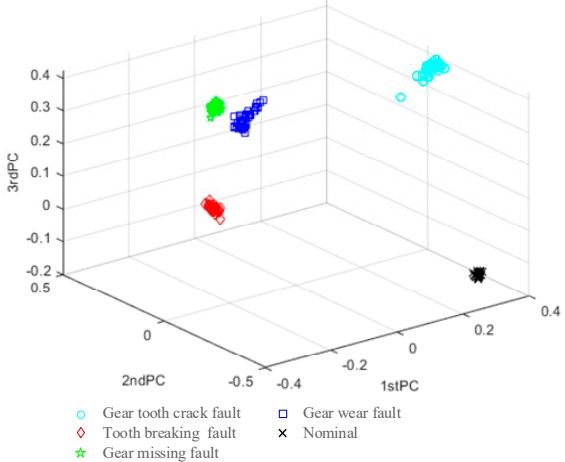

**Figure 13.** Clustering result of wavelet entropy under condition 3.

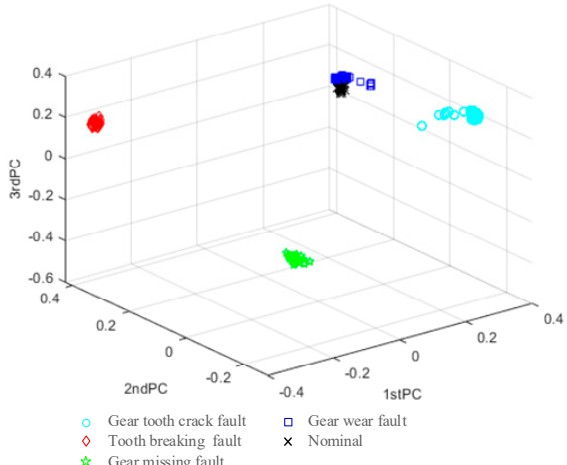

**Figure 14.** Clustering result of wavelet entropy under condition 4.

As shown in Figures 11–14, when the feature vector extracted by the wavelet entropy method was used to identify the degree of fault type, a certain degree of aliasing appeared under the operation condition 1 and operation condition 4. Therefore, the method proposed in this paper is desirable and has better clustering results than does the wavelet entropy method in fault feature extraction.

5.2.2. Comparison between the Proposed Method with VMD-Time-Frequency Information Entropy

Variational mode decomposition (VMD) is a completely non-recursive, adaptive signal processing method proposed by Dragomiretskiy, K. [36]. The method has higher decomposition accuracy and can effectively filter out the white noise in the signal. The specific algorithm steps are as follows:

Step 1: Preprocess the vibration signal.

Step 2: Decompose the signal into six IMFs using the VMD method.

Step 3: Calculate the time-frequency information entropy of each IMF and obtain the 6-dimensional feature vector. Similarly, the feature vector is reduced by PCA. The clustering results are shown in Figures 15–18.

As can be seen from Figures 15–18, the VMD-time-frequency information entropy method also has the phenomenon of fault mixing under the operation condition 1 and the operation condition 4. By contrast, it can be proved that the method proposed in this paper has better performance than the VMD-time-frequency information entropy method in fault feature extraction.

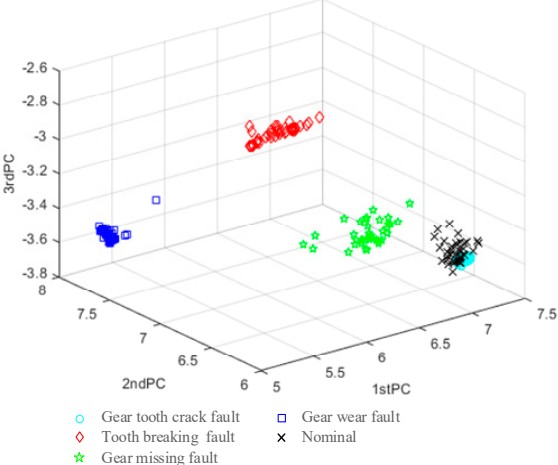

**Figure 15.** Clustering result of variational mode decomposition (VMD)-time-frequency information entropy under condition 1.

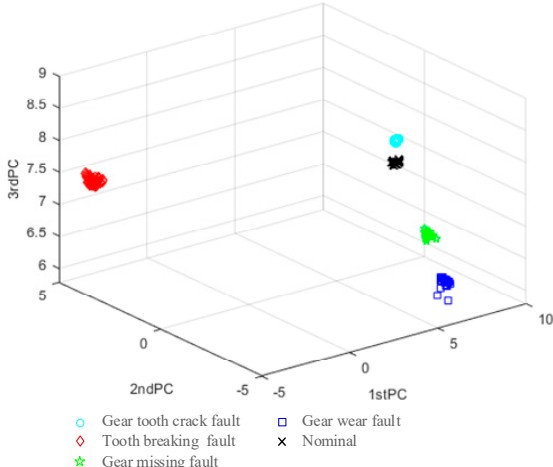

**Figure 16.** Clustering result of VMD-time-frequency information entropy under condition 2.

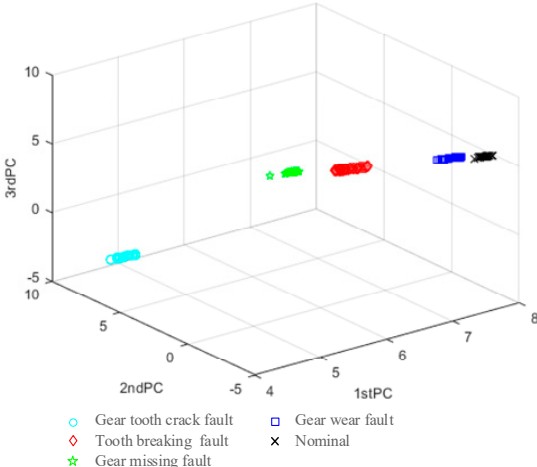

**Figure 17.** Clustering result of VMD-time-frequency information entropy under condition 3.

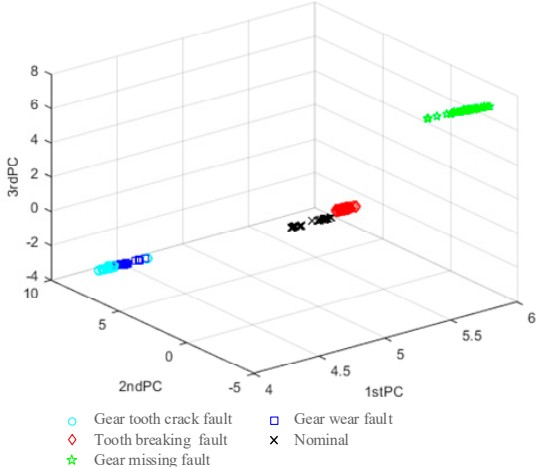

**Figure 18.** Clustering result of VMD-time-frequency information entropy under condition 4.

### 5.2.3. Comparison under Variable Operation Conditions

In order to further demonstrate the advantages of the proposed algorithm, the diagnostic performance of the three methods were compared under variable operation conditions. In this paper, the data under variable operation conditions were spliced together. The specific information is shown in Table 3. We compared the three algorithms mentioned in this paper. The results are shown in Figures 19–21. In total, 128,000 sampling points were extracted from each operation condition, and each fault consisted of a total of 384,000 sampling points under three operation conditions.

**Table 3.** Multi-operation condition data.

| Failure Mode | Variable Operation Conditions (Arranged According to the Order of Operation Conditions) |
|---|---|
| Gear tooth crack fault | condition 1–condition 5–condition 9 |
| Tooth breaking fault | condition 2–condition 4–condition 7 |
| Gear breaking fault | condition 3–condition 5–condition 9 |
| Gear wear fault | condition 12–condition 11–condition 10 |
| Normal | condition 1–condition 6–condition 10 |

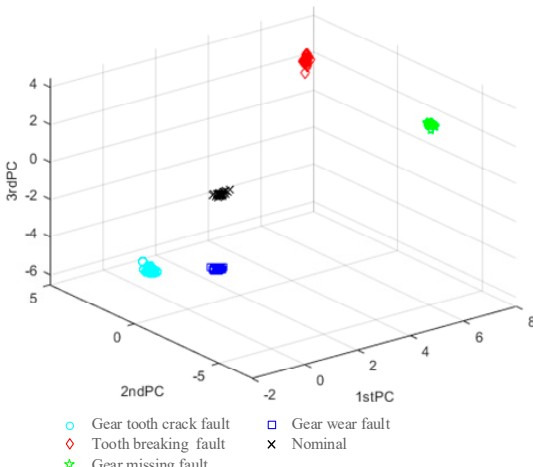

**Figure 19.** Clustering result of ICEEMD-time-frequency information entropy under variable conditions.

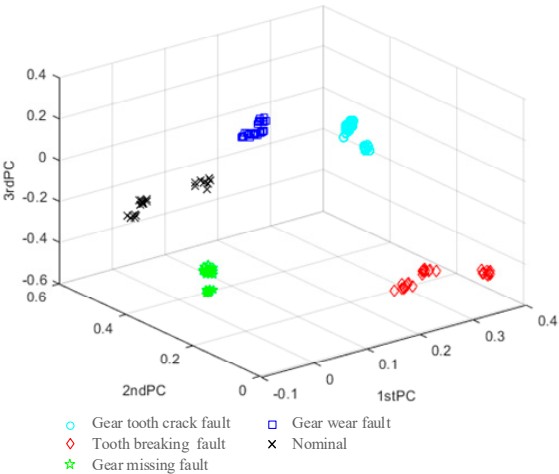

**Figure 20.** Clustering result of wavelet entropy under variable conditions.

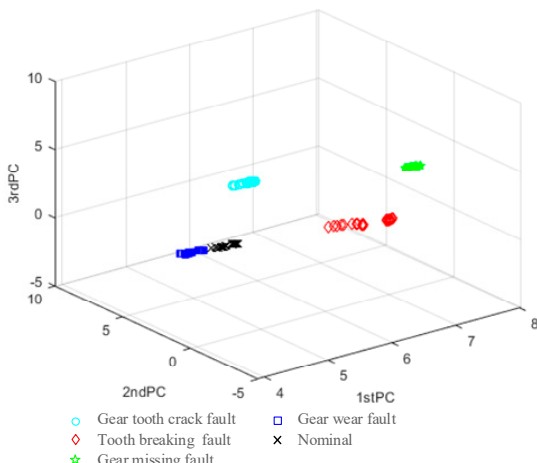

**Figure 21.** Clustering result of VMD-time-frequency information entropy under variable conditions.

Under the variable operation conditions, the algorithm proposed in this paper had obvious clustering of fault features and no fault mixing. However, the wavelet entropy algorithm and VMD-time-frequency information entropy algorithm had a certain degree of mixing for the same fault under

different operation conditions, and the degree of polymerization was not ideal. Therefore, neither of these methods can be applied to the fault diagnosis of variable operation conditions.

### 5.3. Fault Classification of Planetary Gearbox Based on VPMCD

Under a single operation condition, 100 groups of data were selected as the VPMCD training set, and the remaining 100 groups of data were used as the VPMCD test set. Similarly, under variable operation conditions, there were 75 groups of data used as the training set and the remaining 75 groups of data as the test set. The feature vector after PCA dimension reduction was used as the input of VPMCD, and the fault type was used as the output of VPMCD. The corresponding relationship between the number and the fault type is shown in Table 4.

**Table 4.** Corresponding information between number and fault type.

| Number | Fault Type |
| --- | --- |
| 1 | Gear tooth crack fault |
| 2 | Tooth breaking fault |
| 3 | Gear breaking fault |
| 4 | Gear wear fault |
| 5 | Normal |

The classification results are shown in Figures 22–26. It can be seen from the classification results that the diagnostic accuracy rate reached 100% under the single operation condition and variable operation conditions.

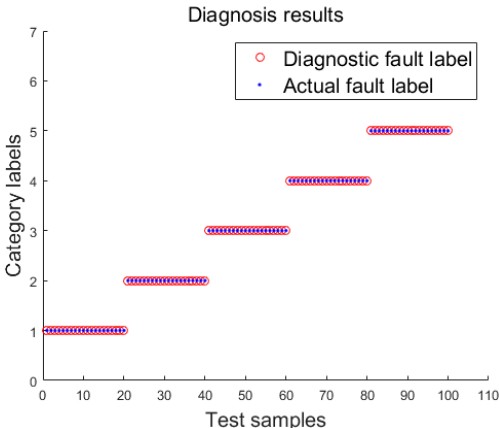

**Figure 22.** Classification result under condition 1.

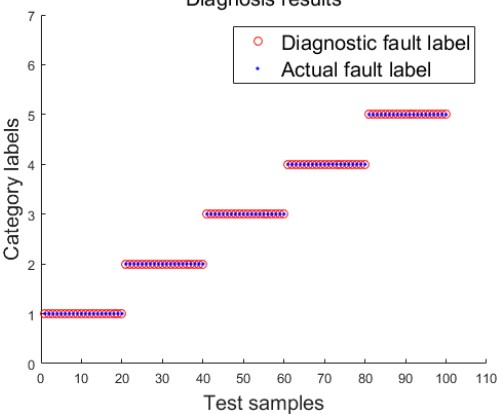

**Figure 23.** Classification result under condition 2.

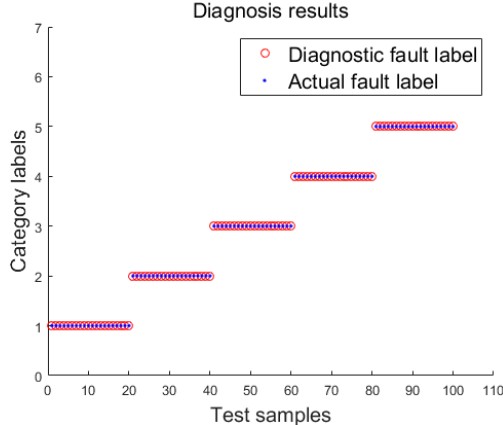

**Figure 24.** Classification result under condition 3.

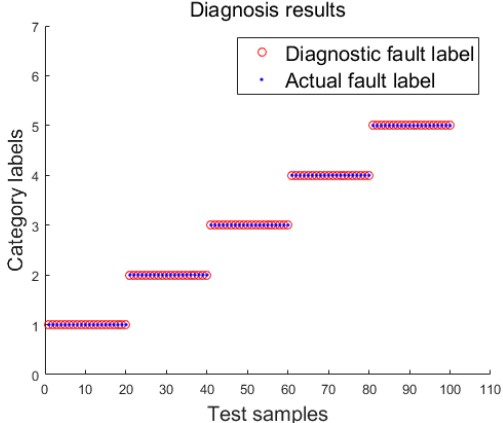

**Figure 25.** Classification result under condition 4.

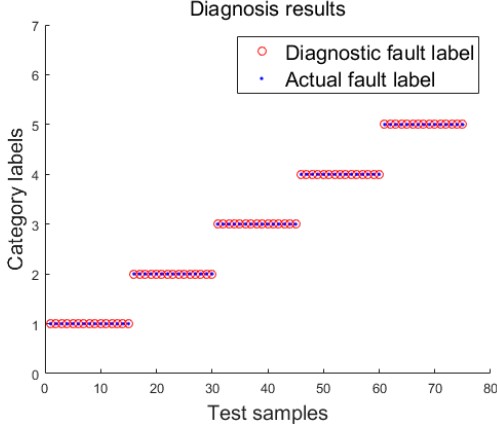

**Figure 26.** Classification result under variable conditions.

In order to reduce the randomness of the diagnosis, we selected 75 groups of data randomly as training sets and the remaining 75 groups of data as test sets. After 100 random classifications, the average classification accuracy of VPMCD was 100% and no testing sample wase misclassified.

## 6. Conclusions

The effective fault diagnosis of planetary gearboxes is indeed difficult and has always attracted the attention of researchers. The weak fault features are difficult to extract from a planetary gearbox because the signal components are nonlinear, nonstationary, and easily drowned out by noise. In this

paper, we propose a fault diagnosis method based on ICEEMD-time-frequency information entropy. The effectiveness of the method is verified by a case study where the proposed method outperforms the wavelet entropy method and the VMD-time-frequency information entropy method in extracting critical fault features. In regard to fault classification, the VPMCD algorithm is adopted to comprehensively consider the correlation about eigenvalues, which can effectively identify the characteristic information of various types of small samples. Moreover, experimental results show that the proposed method can not only accurately diagnose faults under multiple operating conditions, but also produces satisfactory diagnostic performance under variable operating conditions. Therefore, the ICEEMD-time-frequency information entropy and VPMCD method, with favorable robustness and diagnostic performance, has wide applicability in other similar fault diagnoses of rotating machinery. However, to some extent, the performance of the proposed method is limited by computer resources. Future work will focus on optimization of the algorithm to improve its performance.

**Author Contributions:** Conceptualization, Y.W.; Data curation, Y.W.; Methodology, Y.W., H.L., and Z.F.; Writing original draft, Y.W.; Writing review and editing, X.G. All authors have read and agreed to the published version of the manuscript.

**Funding:** This research was funded by Key R & D projects of Shanxi province (No. 2019ZDLGY17-06), the Key Projects for Quality Reliability Design, and Analysis Technology Breakthrough of MIIT under Tender (No. TC190A4DA/6).

**Conflicts of Interest:** The authors declare no conflict of interest.

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
