# Peer review of "Planetary Gearbox Fault Diagnosis Based on ICEEMD-Time-Frequency Information Entropy and VPMCD"

_applsci, doi:10.3390/app10186376_

Round 1

Reviewer 1 Report

The topic is interesting but is not well motivated and discussed. In addition, the innovation and contribution of the manuscript are not clear. The manuscript needs to be revised before being accepted.

  1. In the introduction, the author needs to explain why permutation entropy is used to remove background noise.
  2. A combination of multi-scale permutation entropy and CEEMD is named ICEEMD. The authors are suggested to give reasons and prove that ICEEMD is better than CEEMD through simulation.
  3. In the introduction part, authors are suggested to clearly sort out the logic of the manuscript and the logical relationship to make the contribution and innovation of the manuscript clear. The introduction part of the current manuscript is really chaotic.
  4. In the experimental part, when performing fault feature classification, the manuscript only gives one experiment with randomness. The authors are better to do more experiments and take the average value as the experimental result.

Reviewer 2 Report

The article presents a new practical method of diagnosis of the planetary gearbox. The authors carried out and described very detailed bench research, which deserves recognition. The introduction and review of the literature are well developed. The drawings are clearly and correctly described in the text. Therefore, I believe the article should be published in Applied Sciences. 

Reviewer 3 Report

In this paper, the Authors develop a fault diagnosis method of planetary gearbox based on Improved Complementary Ensemble Empirical Mode Decomposition (ICEEMD)-Time frequency information entropy and Variable Predictive Model-based Class Discriminate (VPMCD).

The method proposed can eliminate noise based on each intrinsic mode function (IMF).

The results provide evidence that the proposed method is superior to the wavelet entropy method and variational mode decomposition (VMD)-time frequency information entropy.

I found that this paper is very interesting and that the obtained results are very promising, however, to further improve I would only recommend improving the introduction and the conclusions and more references on the background (I suggest: doi:10.3390/s20082433, doi:10.3390/machines6030036, doi:10.3390/app10030932, doi:10.3390/machines6040044, doi:10.3390/e20110850). Please cancel from line n. 433 to n.444.

Round 2

Reviewer 1 Report

Although the authors have addressed all the comments in the response file, no improvement has been made as mentioned in the refresh version of the manuscript.  Moreover, it is better to modify the English further to improve the quality of the manuscript.

Besides, careful proofreading is required as there are many Spelling errors and formatting errors. Such as: the errors in the reference 35,36,37,38,39,40:

(35. Author 1, A.; Author 2, B. Book Title, 3rd ed.; Publisher: Publisher Location, Country, 2008; pp. 154–196. 433
36. Author 1, A.B.; Author 2, C. Title of Unpublished Work. Abbreviated Journal Name stage of publication 434 (under review; accepted; in press). 435
37. Author 1, A.B. (University, City, State, Country); Author 2, C. (Institute, City, State, Country). Personal 436 communication, 2012. 437
38. Author 1, A.B.; Author 2, C.D.; Author 3, E.F. Title of Presentation. In Title of the Collected Work (if 438 available), Proceedings of the Name of the Conference, Location of Conference, Country, Date of 439 Conference; Editor 1, Editor 2, Eds. (if available); Publisher: City, Country, Year (if available); Abstract 440 Number (optional), Pagination (optional). 441
39. Author 1, A.B. Title of Thesis. Level of Thesis, Degree-Granting University, Location of University, Date of 442 Completion. 443
40. Title of Site. Available online: URL (accessed on Day Month Year).)

These problems are caused by the Authors contempt for this paper. 

Author Response

Point 1: Although the authors have addressed all the comments in the response file, no improvement has been made as mentioned in the refresh version of the manuscript.

Response 1: We quite appreciate your favorite consideration and insightful comments concerning our manuscript entitled “Planetary gearbox fault diagnosis based on ICEEMD-Time frequency information Entropy and VPMCD”. I am very sorry that due to my negligence, I only uploaded the modification manuscript of word version and did not update the PDF version, which caused some misunderstandings. In the new version of the manuscript, I have been refreshed the manuscript of PDF version to be consistent with the content of Word version. We hope this revision can meet with approval. The revised portions are marked in red underline and highlighted with the yellow background. The main revisions corresponding to the reviewers’ comments are as follows:

  • In the introduction, the author needs to explain why permutation entropy is used to remove background noise.

In this paper, the Multi-scale permutation entropy is used to remove background noise. Multi-scale permutation entropy is a powerful nonlinear characteristic parameter, with good consistency and stability, and can describe signals on multiple scales. Some results has been indicated that the multi-scale permutation entropy method has a good separability between signal and noise components. The reason of using Multi-scale permutation entropy has been added, which is located in line 61- 68 highlighted with the yellow background and marked in red underline.

  • A combination of multi-scale permutation entropy and CEEMD is named ICEEMD. The authors are suggested to give reasons and prove that ICEEMD is better than CEEMD through simulation.

The method proposed can eliminate noise based on each intrinsic mode function (IMF). ICEEMD can better remove noise than CEEMD and retain signal characteristics. The new results of comparison between ICEEMD and CEEMD have been added, which is located in line 152 -174 highlighted with the yellow background and marked in red underline.

  • In the introduction part, authors are suggested to clearly sort out the logic of the manuscript and the logical relationship to make the contribution and innovation of the manuscript clear. The introduction part of the current manuscript is really chaotic.

We sort out the logic of the manuscript and the logical relationship,which can be found in line 34 - 44, and in line 80-91 highlighted with the yellow background and marked in red underline.

  • In the experimental part, when performing fault feature classification, the manuscript only gives one experiment with randomness. The authors are better to do more experiments and take the average value as the experimental result.

The fault feature matrix after dimensionality reduction is classified by VPMCD when performing fault feature classification. In order to reduce the randomness of the diagnosis, we select 75 groups of data randomly as training sets and the remaining 75 groups of data as test sets. After 100 times of random classification, the average classification accuracy of VPMCD is 100.00% and no testing sample is misclassified, which is located in line 359- 361 highlighted with the yellow background and marked in red underline. In the future, we will do more experiments to verify the effectiveness of the fault feature classification method.

Point 2: It is better to modify the English further to improve the quality of the manuscript.

Response 2: Thanks for your comment. We have revised the manuscript carefully and the corresponding revisions of manuscript, which can be found in line 52 - 61, and in line 362-378 highlighted with the yellow background.

Point 3: Careful proofreading is required as there are many Spelling errors and formatting errors. Such as: the errors in the reference 35,36,37,38,39,40:

Response 3: Thanks for your comment. Firstly, we cancel from line 433 to 444. In addition, we add more references on the background, which is located in line 389-400 highlighted with the yellow background and marked in red underline. Finally, I have spent some time on the spell and grammar and correct some errors in the paper.

Round 3

Reviewer 1 Report

The authors have addressed all the comments and done the required revisions properly. The paper should be suitable for publication and be of interest to the audience.